# Adherence to a Healthy Diet and Risk of Multiple Carotid Atherosclerosis Subtypes: Insights from the China MJ Health Check-Up Cohort

**DOI:** 10.3390/nu16142338

**Published:** 2024-07-19

**Authors:** Jingzhu Fu, Yuhan Deng, Yuan Ma, Sailimai Man, Xiaochen Yang, Canqing Yu, Jun Lv, Hui Liu, Bo Wang, Liming Li

**Affiliations:** 1Department of Epidemiology and Biostatistics, School of Public Health, Peking University, Beijing 100191, China; fujingzhu2016@bjmu.edu.cn (J.F.); 2316391136@bjmu.edu.cn (S.M.); yucanqing@pku.edu.cn (C.Y.); lvjun@bjmu.edu.cn (J.L.); 2Peking University Health Science Center, Meinian Public Health Institute, Beijing 100191, China; paul@meinianresearch.com; 3Key Laboratory of Epidemiology of Major Diseases (Peking University), Ministry of Education, Beijing 100191, China; 4Meinian Institute of Health, Beijing 100083, China; 2011210107@stu.pku.edu.cn (Y.D.); yuan.ma@meinianresearch.com (Y.M.); 2211110242@stu.pku.edu.cn (X.Y.); 5Department of Social Medicine and Health Education, School of Public Health, Peking University, Beijing 100191, China; 6Chongqing Research Institute of Big Data, Peking University, Chongqing 400000, China; 7School of Population Medicine and Public Health, Chinese Academy of Medical Sciences & Peking Union Medical College, Beijing 100730, China; 8Peking University Center for Public Health and Epidemic Preparedness & Response, Beijing 100191, China; 9Institute of Medical Information, Chinese Academy of Medical Sciences & Peking Union Medical College, Beijing 100020, China

**Keywords:** prospective cohort study, diet, carotid atherosclerosis, nutrition, China

## Abstract

Aim: Early-stage phenotypes of carotid atherosclerosis (CAS), such as increased carotid intima-media thickness (cIMT), and advanced-stage phenotypes, such as carotid plaque (CP), are at risk for adverse ischemic stroke events. There is limited evidence regarding the causal association between dietary patterns and the risk of CAS in Chinese adults. We therefore examined multiple dietary patterns associated with the risk of CAS and identified the optimal dietary pattern for preventing CAS. Methods: We analyzed data collected from the prospective MJ Health Check-up Study (2004–2020), including 13,989 participants 18–80 years of age without CAS. The dietary intake was measured using validated food frequency questionnaires, and dietary pattern scores were calculated for four a priori and four a posteriori dietary patterns. The Cox model was used to estimate the adjusted hazard ratios (HRs) relating various dietary pattern scores to the risk of CAS. Results: During 43,903.4 person-years of follow-up, 3732 incidents of increased cIMT and 2861 incident CP events were documented. Overall, the seven dietary patterns, except for the high-protein diet, exhibited significant associations with the risk of increased cIMT and CP. Comparing the highest and lowest quartiles, the a posteriori high-fiber dietary pattern (HFIDP) score demonstrated the strongest inverse associations with the risk of increased cIMT (HR 0.65 [95% confidence interval (CI) 0.59–0.71]) and CP (HR 0.65 [95% CI 0.59–0.73]); conversely, another a posteriori high-fat dietary pattern (HFADP; i.e., incorporating high-fat and processed foods) demonstrated the strongest positive associations with the risk of increased cIMT (HR 1.96 [95% CI 1.75–2.20]) and CP (HR 1.83 [95% CI 1.61–2.08]) (all *p* for trend < 0.01). Conclusions: Multiple dietary patterns are significantly associated with the risk of early- and advanced-stage phenotypes of CAS. Notably, a high adherence to an HFIDP and low adherence to an HFADP may confer the greatest risk reduction for CAS.

## 1. Introduction

Against the backdrop of global population aging, cardiovascular diseases (CVDs) will continue to receive widespread attention in the coming decades [1]. Carotid atherosclerosis (CAS) is not only widely recognized as an etiology and risk factor for ischemic stroke, but has also been found to be a strong predictor of coronary artery events and the prognosis of CVD [2]. It remains latent and asymptomatic for an extended period from the early-stage phenotype (i.e., increased carotid intima-media thickness (cIMT)) to the advanced-stage phenotype (i.e., carotid plaque (CP)) [3]. It has been reported that these two phenotypes of CAS affect one-quarter to one-third of the adult population globally, including in China [4,5]. Although surgery and pharmacological treatments have been used as effective interventions for CAS, they are expensive and may led to adverse effects. Therefore, it is critical to identify risk factors for CAS and implement early intervention.

Accumulating data support dietary changes as a potentially effective strategy for preventing CAS [6]. In recent years, an increasing number of studies have gradually shifted from the evaluation of single dietary factors to that of the overall diet, exploring its association with health outcomes [7]. Dietary patterns that encompass a diverse range of foods, nutrients, and beverages offer the advantage of favorable public health implications because they reflect the overall impact of the diet and aid in providing dietary guidance. Previous dietary pattern scores, such as the Dietary Approaches to Stop Hypertension (DASH), Mediterranean diet score, and healthful plant-based diet index (hPDI), have been found to be associated with cardiovascular health [8,9]. Globally, the Prospective Urban Rural Epidemiology healthy diet (PURE) has been shown to provide more health benefits in the prevention of CVD than the DASH and Mediterranean diet [8]. The key to this healthy diet is to include diverse natural foods in moderation, rather than restricting intake to a small number of food categories. A recent meta-analysis presented inconsistent recommendations that a vegetarian diet was associated with improved arterial stiffness and reduced atherosclerosis compared to an omnivorous diet in healthy individuals [10]. Moreover, an accumulating body of observational research has identified both a priori (e.g., dietary inflammation index (DII) and dietary diversity score (DDS)) [11,12] and a posteriori (e.g., “prudent” and “convenience”) [13] dietary patterns associated with CAS in Western populations. Nevertheless, there is little evidence supporting the prospective association between dietary patterns and the risk of CAS in Asian populations [14,15]. Given the heterogeneity in the associations between dietary patterns and CAS across different racial/ethnic populations, it is necessary to identify dietary patterns associated with CAS in Chinese populations and provide relevant dietary guidance. Additionally, these dietary patterns have different scoring methods and may affect CAS differentially. To the best of our knowledge, a comparative analysis investigating the association between dietary patterns and CAS has not been performed.

As such, the present study aimed to assess the associations between four a priori dietary patterns scores—specifically, DDS, DII, PURE, and hPDI—as well as four a posteriori dietary patterns derived from the MJ dietary data, and the onset of different subtypes of CAS in adults to compare the performance of various dietary pattern scores.

## 2. Materials and Methods

### 2.1. Study Population

This study used data from the Beijing MJ Health Screening Center (Beijing, China) obtained between January 2004 and December 2020. Details of the MJ study have been described elsewhere [16]. Briefly, individuals from the general adult population who participated in annual health examinations were included. Follow-up measurements were performed periodically during the annual health examinations. At each visit, data regarding demographics, medical history, diet, and lifestyle were collected using structured questionnaires. Participants underwent anthropometric measurements, thorough medical examinations, and blood and urine investigations. In the present study, participants were followed up regarding CAS until 31 December 2021.

This study included participants who had undergone ≥ 2 health examinations. Among them, those with missing data regarding dietary pattern scores, those with implausible energy intake (i.e., ≤520 kcal/day [minimum energy required for survival]) or ≥8000 kcal/day (approximately 4 times the mean energy intake), those with a baseline history of CAS, CVD, or cancer, and those aged < 18 or >80 years of age at baseline were excluded. The final sample comprised 13,989 participants with complete data (Appendix A). Salient characteristics of individuals included and excluded from the current study were largely comparable. The study protocol was approved by the Institutional Review Board of Peking University (IRB00001052–19077). Written informed consent was obtained from all the participants.

### 2.2. Assessment of Dietary Intakes

In the MJ cohort, participants provided data on their dietary intake during the preceding month using a validated semiquantitative 25-item food frequency questionnaire (FFQ) covering foods, beverages, cooking oil, and salt typically consumed in this population.

Participants were asked to specify their food consumption frequency of specified portion sizes in the FFQ (including five response options: from “<1 serving/week” to “≥2 servings/day”). The definition of one serving size of each food item was given using an example, such as “1 cup is equivalent to 240 mL of milk, 240 mL of yogurt, and four tablespoons of powdered milk”. Daily alcohol consumption was calculated as grams of pure alcohol according to alcohol type, amount consumed, and frequency. Total energy and nutrient intakes were computed from the consumption frequencies and portion sizes, using the 2009 Chinese Food Composition Table [16]. The reliability and validity of the MJ Health Screening Center’s FFQ have been evaluated previously [17,18]. Regarding validity, the Spearman rank correlation coefficients for food groups as measured by the FFQ and intakes of the same foods assessed by a 4-day dietary record ranged from 0.29 to 0.47 (all *p*-values < 0.0001).

### 2.3. Assessment of Dietary Patterns

To better capture the dietary features of our study population, we scored every participant according to their adherence to specific dietary patterns, including the four a priori dietary patterns (i.e., DDS, DII, hPDI, and PURE), as well as four a posteriori dietary patterns derived from MJ data. The calculation of a priori dietary patterns is detailed in Appendix A [8,19,20,21,22,23,24]. 

To determine the major a posteriori dietary patterns, principal component factor analysis was used with varimax rotation based on the same food group data transformed to z scores. Four major dietary patterns were identified through a comprehensive consideration of eigenvalues, explained variance, scree plot, and interpretability (see more details in Appendix A [25,26]). For each dietary pattern, factor scores were calculated for all participants by summing the standardized intake of food groups weighted according to their factor loadings.

## 3. Covariates

We obtained covariate information on participants’ demographic factors, lifestyles, medical history, family history of CVD, and medication use from the baseline questionnaire. Five lifestyle factors were considered to define a low-risk lifestyle, namely, smoking, alcohol consumption, physical activity (PA), body mass index (BMI), and waist-to-hip ratio (WHR), according to previous studies [27]. The details of the low-risk group definition for each lifestyle are described in Appendix A Section S3 [27,28,29,30,31,32]. The physical assessments included standardized measurements of weight, height, waist circumference (WC), hip circumference, and blood pressure. The blood indicators were measured using blood samples from participants who had fasted for at least 8 h. A detailed assessment of the covariates is described in Appendix A [33,34].

## 4. Ascertainment of CAS

Trained sonographers utilized a Doppler ultrasound system (Sonoscape S50, SonoScape Medical Corp., Shenzhen, China) equipped with a 7.5 MHz high-resolution linear array transducer to conduct carotid ultrasonography. The participants were required to remain in a supine position with their heads turned 45° to the contralateral side of the artery. The measurements were conducted over a minimum of a 10 mm length on both sides at the far wall in the common carotid artery, longitudinal and perpendicular to the ultrasound beam, in the lateral view, at least 5 mm proximal to the bifurcation in an area with clearly defined lumen-intima and in a region free of plaque. The cIMT was calculated as the distance from the edge of the first echogenic line to the edge of the second echogenic line. An increased cIMT was defined as being 1.0 mm or greater [35].

The procedure for detecting plaques involved scanning the near and far walls of the common carotid artery, the carotid bifurcation, the external carotid artery, and the internal carotid artery. A plaque was defined according to the latest version of the Mannheim Carotid Intima-Media Thickness and Plaque Consensus: a discrete cIMT of 1.5 mm or larger or focal thickening of 0.5 mm or larger or 50% greater than the surrounding cIMT in any of the aforementioned arterial segments [36].

## 5. Statistical Analysis

All statistical analyses were performed with R version 4.0.3 (R Foundation for Statistical Computing, Vienna, Austria) or SAS 9.4 edition for Windows (SAS Institute, Inc., Cary, NC, USA). The normality of the continuous variables was examined by the Kolmogorov–Smirnov test. We performed log transformations on continuous variables with skewed distributions. The age-adjusted baseline characteristics of the participants were presented using the geometric mean (95% confidence interval [CI]) for the continuous variables and a percentage for the categorical variables. As per the STORBE guidelines [37,38], we did not use *p* values to evaluate the differences in the participants’ baseline characteristics. The mean score of 8 dietary patterns and the mean intake of 16 major food/drink groups between sexes or different lifestyles were estimated and compared using a generalized linear model. To characterize the dietary profiles, the mean intake of foods (g per day) and nutrients (% of energy) were calculated according to the quartile group (lowest and highest) of the a priori dietary score and the factor loadings for the a posteriori dietary patterns. The food and nutrient intake of the individuals in the lowest and highest quartile groups of the a priori diet score characterized the dietary profile of the least or most healthy diets.

Cox proportional hazards regression models were applied to estimate the HR and corresponding 95% confidence interval (CI) for the association of the various dietary pattern scores with the risk of CAS (i.e., increased cIMT and CP), comparing the highest to lowest diet quartiles. The proportional hazards assumption was verified using Schoenfeld residuals, and the results indicated that the assumptions were not violated. The person-years of follow-up were calculated from baseline to the date of CAS diagnosis, loss to follow-up, or termination of follow-up (31 December 2021). To conceptualize the model and identify a minimally sufficient adjustment set, a directed acyclic graph (DAG) derived from the current literature, which was based on the protocol of “Evidence Synthesis for Constructing DAGs” was constructed [39] (Appendix A). Based on the DAG, the final models were adjusted for the following: age; sex; ethnicity; marital status; educational attainment; personal annual income; smoking status; drinking (i.e., alcohol) status; PA; total energy intake; multivitamin supplement use; BMI; individual history of diabetes, hypertension and/or dyslipidemia; and family history of CVD. To test for linear trends, the median value of the diet scores was assigned within each quartile and modeled as a continuous variable. From a public health perspective, the population attributable risk (PAR)—which is the percentage of incident CAS that would be avoided in a population during follow-up if all the participants had adopted a healthier diet—was calculated [40]. Restricted cubic spline plots with five knots were used to explore the dose–response association between the diet scores and risk of CAS. Stratified analyses according to pre-specified subgroups (i.e., different sex and lifestyle risk groups) were also performed using Cox regression models, and the significance of the potential effect modification was tested by including multiplicative interaction terms in the model, where observed, and presented as joint analyses. Joint analyses of sex or lifestyles with various diet scores were performed to examine how their combination was associated with the risk of CAS. To determine which diet scores were most strongly associated with CAS, a receiver operating characteristic (ROC) analysis was performed to assess the predictive ability of the different diet scores [41]. The predictive performance of each Cox model was measured according to the area under the ROC curve (AUC). The significance level of the differences in the AUC values was calculated using the DeLong test [42]. To examine the mediation effects of metabolic syndrome (MetS; Yes or No), Cox regression-based mediation analyses were used to decompose the total effects of the dietary patterns on CAS into natural direct and indirect effects through MetS, and to calculate the proportion of mediation (PM) accordingly [43]. Linear regressions were used to further analyze the associations between the dietary patterns and MetS components. In the sensitivity analyses, the effect of removing potential mediators (BMI, WC, diabetes, dyslipidemia, and hypertension) was tested on the estimates of the final models. Furthermore, we also excluded individuals with new-onset CAS in the first year of follow-up to observe the stability of the results. Due to the potential for type I error from multiple comparisons, the findings from the secondary and subgroup analyses should be interpreted as exploratory. A two-sided *p*-value of <0.05 was considered statistically significant.

## 6. Results

### 6.1. Population Characteristics

The mean (± SD) age of the 13,989 individuals included in the sample was 42.5 ± 8.53 years, 60.9% of the participants were males, and 70.3% adhered to healthy lifestyles. The age-adjusted baseline characteristics according to the lowest and highest quartiles of the various dietary scores are summarized in Table 1. The median (interquartile range [IQR; P25, P75]) HFIDP score, HFADP score, high-protein dietary pattern (HPDP) score, high-carbohydrate dietary pattern (HCDP) score, DDS, DII, PURE score, and hPDI were −0.32 (−0.58–0.44), −0.20 (−0.69–0.48), −0.096 (−0.69–0.57), −0.050 (−0.61–0.49), 2.00 (2.00–3.00), −0.047 (−2.30–2.19), 1.00 (0.00–2.00), and 49.0 (47.0–51.0), respectively. The participants with a higher adherence to healthy diets (i.e., HFIDP, DDS, PURE, and hPDI) shared similar characteristics. Specifically, they were more likely to be female, had healthier lifestyles (i.e., no smoking, limited alcohol consumption, regular PA, and moderate BMI and WHR), consumed more vegetables, fruits, and whole grains, and were less likely to have diabetes, dyslipidemia, and MetS. In contrast, the participants with a higher adherence to unhealthy diets (i.e., DII or HFADP) generally tended to be male and had dyslipidemia, MetS, a lower consumption of vegetables, fruits, whole grains, and multivitamin supplement intake, and unhealthier lifestyles.

The components of the various dietary patterns are summarized in Table 2. Based on the mean intake of foods according to the level of the a priori diet score, all the “most healthy” dietary patterns (i.e., the highest or lowest quartile of dietary scores) were associated with a higher intake of legumes, light and dark vegetables, fruits, whole grains, and tubers. However, dairy products, eggs, meat, seafood, offal, sweet snacks, and sugar-sweetened beverages all exhibited a higher intake in the “most healthy” dietary pattern groups of DDS, PURE, and DII, but exhibited the opposite trend in the hPDI. This corresponds to the differences in each nutrient across the dietary patterns, with the “most healthy” dietary pattern groups of DDS, PURE, and DII exhibiting a higher energy intake from protein and polyunsaturated fat, and a higher intake of cholesterol, while hPDI exhibited the opposite trend.

The mean of the eight dietary scores and major food components for different sexes and lifestyle statuses are reported in Figure 1. Significantly higher mean HPDP and HFADP scores and intakes of some food groups (i.e., meat, processed foods, fried foods, sugar-sweetened beverages, refined grains, legumes, offal, and seafood) were found in the males, while the females had significantly higher HFIDP, HCDP, DDS, hPDI, and PURE scores and intakes of other food groups (i.e., dairy, sweet snacks, tubers, whole grains, fruits, dark and light vegetables; all *p* < 0.05; Figure 1A,B). Compared to the participants with healthy lifestyles, those with unhealthy lifestyles exhibited significantly higher DII and HFADP scores and intakes of some food groups (i.e., meat, processed foods, fried foods, sugar-sweetened beverages, refined grains, offal, and seafood), whereas the intake of other food groups was the opposite (all *p* < 0.05; Figure 1C,D).

### 6.2. Association between Dietary Patterns and CAS

During a median of 2.24 years (43,903.4 person-years), 3732 incidents of increased cIMT and 2861 incident CP events were observed. Except for the HPDP, all other dietary patterns were associated with a risk of increased cIMT and CP (Table 3). Compared with the participants with the lowest quartile intake, the adjusted HRs (95% CIs) of those with the highest quartile intake of an HFADP, HCDP, and DII were 1.96 (1.75–2.20), 1.17 (1.07–1.29), and 1.66 (1.48–1.87) for increased cIMT, and 1.84 (1.61–2.10), 1.20 (1.08–1.34), and 1.59 (1.39–1.82) for CP, respectively. The adjusted HRs (95% CIs) of those with the highest quartile intake of an HFIDP, DDS, PURE, and hPDI were 0.65 (0.59–0.71), 0.74 (0.63–0.87), 0.76 (0.67–0.87), and 0.78 (0.71–0.85) for increased cIMT, and 0.65 (0.59–0.73), 0.72 (0.59–0.87), 0.75 (0.65–0.87), and 0.79 (0.72–0.88) for CP, respectively. Similar associations between the dietary patterns and CAS were also observed in this sensitivity analysis (Appendix A). The PAR of increased cIMT and CP was highest for an HFADP. The PAR of increased cIMT and CP in relation to the higher adherence to an HFADP were 48.0% (42.0–53.0%) and 44.0% (38.0–50.0%), respectively, suggesting that approximately one half of the incident CAS in this population during the follow-up period may have been prevented if all the participants had displayed the lowest adherence to an HFADP. The restricted cubic splines exhibited significant non-linearity for associations between the energy-adjusted a posteriori dietary patterns (i.e., HFIDP, HFADP, and HPDP; *p* < 0.01 for non-linearity) and a priori dietary patterns (i.e., DDS, DII, and hPDI; *p* < 0.05 for non-linearity) and the risk of increased cIMT and CP (Figure 2 and Appendix A).

The results of the analysis stratified according to sex and lifestyles are presented in Figure 3 and Appendix A. Multiplicative interactions were found between the DDS and composite lifestyle scores (*p*-interaction < 0.05). In the stratified analyses, a negative association between the DDS and CP risk was observed among the participants who adhered to healthy lifestyles (Q4 versus [vs.] Q1, HR 0.79 [95% CI 0.62–1.00]) but not unhealthy lifestyles. However, the DDS was significantly and inversely associated with the risk of increased cIMT, regardless of the composite lifestyle status (*p* < 0.05). The synergistic effect of lifestyle with the DDS on CAS (i.e., increased cIMT and CP) is shown in Appendix A. For instance, from the participants with the highest adherence to the DDS (Q4) to the highest adherence to the DDS (Q4) and healthy lifestyles, the HR (95% CI) for increased cIMT and CP increased from 0.77 (0.66–0.91) and 0.77 (0.64–0.92) to 0.66 (0.60–0.74) and 0.64 (0.57–0.72), respectively.

### 6.3. Comparison of Dietary Scores and Mediation Analysis

Among all the dietary patterns, the HFIDP and HFADP exhibited a stronger association with the risk of CAS (i.e., increased cIMT and CP) than the other dietary patterns (Table 3). Similarly, in the tests comparing the differences in the AUC values between the diet scores, the HFIDP and HFADP scores exhibited significantly stronger associations with the risk of CAS than the other dietary patterns (Table 4). Moreover, the HFIDP score was the most similar to the HFADP score, with only slightly larger AUC values observed for the HFIDP score, which were not statistically significant. 

For simplicity, only the results of the mediation analysis of MetS between an HFIDP/HFADP and increased cIMT/CP are reported. By decomposing the total associations into natural direct and indirect effects, the PM by MetS differed slightly between increased cIMT and CP and remained relatively stable for increased cIMT or CP, regardless of the dietary pattern type (Table 5). Regarding the associations with increased cIMT, the PM was 3.70% for the HFIDP and 4.41% for the HFADP. Regarding the associations with CP, the PM increased to 3.88% for the HFIDP and 5.38% for the HFADP. Furthermore, the association between the HFIDP and HFADP with the MetS component levels (i.e., blood pressure, plasma lipids and glucose levels, and WC) were examined in the overall population (Appendix A). A significantly positive association was verified between an HFADP and the TG and WC levels (*p* for trend < 0.01) and a significantly inverse association with the HDL-C levels (*p* for trend = 0.017) after adjusting for the covariates. Nevertheless, it was also found that the HFIDP was negatively associated with the DBP and WC, although the association was not statistically significant. 

## 7. Discussion

This large prospective cohort study of Chinese adults observed and compared eight dietary patterns based on different mechanisms to identify the optimal dietary pattern to reduce the risk of CAS from various perspectives. In general, while the DDS, DII, hPDI, PURE, and HCDP scores were significantly associated with the CAS risk (i.e., increased cIMT and CP), the strongest positive and inverse associations were observed between a high adherence to an unhealthy diet (i.e., HFADP) and a high adherence to a healthy diet (i.e., HFIDP) and the risk of CAS. The HFIDP and HFADP scores appeared to be slightly more predictive of CAS than the other dietary scores. Furthermore, not only did MetS partially mediate the association between an HFIDP and HFADP with CAS, but further revealed that the consumption of an HFADP was significantly associated with an unfavorable lipid profile and a higher WC (diagnostic indicators for MetS). Our findings suggest that the key to a healthy diet in preventing CAS is to prioritize plant-based foods, limit high-fat and processed food consumption, and incorporate a variety of natural and anti-inflammatory foods in moderate amounts.

CVD is the leading cause of diet-related deaths worldwide, especially for China [44], and various dietary scores associated with the CVD risk have been identified [8,45]. Several studies have reported that healthy dietary patterns, as assessed by high dietary fiber intake and the hPDI, DDS, PURE DASH, and Mediterranean diet, are associated with a lower risk of incident CVD [7,46,47,48]. Many food items included in the DASH and Mediterranean diet scores, such as olive oil, wine, butter, stick margarine, cheese, pastries, and low-fat/skim milk, are not applicable to Chinese populations and are rarely consumed owing to different dietary cultures and economic levels. Therefore, for the a priori dietary pattern, we used the PURE score, which is similar to the DASH and Mediterranean diet scores, and more closely approximates Chinese dietary habits. The PURE score has been shown to be superior to the DASH and Mediterranean diet scores in relation to CVD [8]. However, there is a paucity of evidence describing a prospective association between dietary patterns and the risk of CAS, especially in Chinese adults [14,49]. A prospective cohort study of 73,890 participants from the United States and a recent meta-analysis addressing relevant research worldwide revealed a protective effect of plant-based dietary patterns, including vegan or vegetarian diets, in preventing cardiovascular events [9,50]. Consistent with this result, we found that a higher adherence to an HFIDP, consisting of a high intake of legumes, vegetables, fruits, and tubers, and the higher adherence to the hPDI, eating more healthy plant foods and fewer unhealthy plant and animal foods, were associated with a reduced risk of developing CAS (i.e., increased cIMT and CP). Both dietary patterns are characterized by high levels of dietary fiber, such as legumes, vegetables, fruits, and tubers. Therefore, the vascular benefit of these two dietary patterns may work through dietary fiber, as dietary fiber intake has been found to be associated with a reduced risk of CVD by reducing inflammation levels [51,52]. Additionally, a recent meta-analysis of cross-sectional studies conducted by Saz-Lara et al. suggested that a vegetarian diet was associated with improved arterial stiffness and reduced atherosclerosis compared to an omnivorous diet in healthy individuals [10]. This also further implies that plant-based dietary patterns and dietary patterns that advocate for dietary diversity may have differing protective effects on the risk of CAS. To our knowledge, a comparative analysis of the associations between these two dietary patterns and the risk of CAS has not been conducted in prospective cohort studies. Our study found that a stronger inverse association between a high adherence to an HFIDP and CAS risk than the DDS and PURE, demonstrating that plant-based diets appeared to be more effective than dietary diversity in reducing the CAS risk. Meanwhile, the associations between an HFIDP and the CAS risk remained robust after adjustment for the DDS and PURE score, suggesting that the HFIDP has unique properties over other dietary indices in evaluating the dietary quality in CAS prevention.

Interestingly, although the HFIDP and hPDI contain a similar composition of healthy plant-based diets, the HFIDP score exhibited markedly stronger associations with CAS than the hPDI score. This may be because the hPDI restricts the consumption of animal-based foods. Animal-based foods, including protein-rich eggs, milk, and seafood, contain antioxidant peptides that ameliorate atherosclerosis [53,54]. Furthermore, both the PURE score and DDS include a wide variety of foods to assess the overall dietary quality. These patterns all have in common an emphasis on increasing the intake of fruits, vegetables, legumes, fish, and dairy, but no restrictions on seafood, eggs, and milk. There is evidence that more diverse food attributes and micronutrient-rich diets, rather than increased quantities of nutrients, contribute to cardiovascular health [12]. Only a cross-sectional study conducted in the Chinese middle-aged population reported that the DDS was inversely associated with the risk of CAS [14]. Similarly, we found that the inverse association between the PURE score and DDS and the risk of CAS remained robust after adjusting for the hPDI, suggesting that the protective effect of the PURE score and DDS on CAS is not only attributed to the inclusion of a healthy plant-based diet, but also underscores the importance of dietary diversity. Therefore, under the premise of advocating a plant-based diet, not restricting the diversity of natural foods may contribute to preventing the occurrence of CAS.

In contrast, poor dietary patterns are associated with increased risks of CVD and CAS. Certain diets have anti- or pro-inflammatory properties and have been shown to play an important role in the development of CVD [55]. The DII, a recognized tool for assessing the inflammatory potential of a diet, has been extensively used to analyze its impact on various chronic diseases [56]. To our knowledge, the present study is the first to link the DII with the incidence of CAS in a Chinese population. We observed that a higher DII was a harmful dietary pattern in this population. This finding was supported by epidemiological studies in Western populations that reported a positive association between a higher inflammatory potential of habitual dietary patterns and CAS and cardiovascular morbidity and mortality [57,58]. Moreover, our study also indicated that the DII score was negatively correlated with the HFIDP (r = −0.75, *p* < 0.001), PURE (r = −0.72, *p* < 0.001), DDS (r = −0.54, *p* < 0.001), and hPDI (r = −0.19, *p* < 0.001), suggesting that a healthy diet may reduce the risk of CAS through anti-inflammatory mechanisms. For example, the aforementioned healthy dietary patterns recommend the increased consumption of vegetables that are rich in anti-inflammatory compounds such as vitamins, flavonoids, carotenoids, and fiber. Offal, sugar-sweetened beverages, and processed and fried foods, as pro-inflammatory foods [59], along with the potential synergistic effects between them [11], may also explain the association between the HFADP and a significantly increased risk of CAS. Previous studies conducted only in Western populations have reported similar results, in that ultra-processed food (UPF) consumption (i.e., sweetened beverages and processed and fried foods) is associated with an increased risk of coronary atherosclerosis [60,61]. Such high-density and low-quality foods not only contain excessive salt, saturated fat, and refined sugars, but also add or generate trans-fatty acids during processing, which may cause the activation of the innate immune system, most likely by an excessive production of pro-inflammatory cytokines associated with the reduced production of anti-inflammatory cytokines [62,63,64]. In addition, a high-fat diet leads to endothelial activation, as evidenced by increased concentrations of the adhesion molecules VCAM-1 (vascular cell adhesion molecule-1) and ICAM-1 (intercellular adhesion molecule-1), in association with raised plasma concentrations of interleukin-6 (IL-6) and tumor necrosis factor-α (TNF-α) [65]. The high-fat diet also increases the circulating levels of interleukin-18 (IL-18), a pro-inflammatory cytokine supposed to be involved in plaque destabilization, associated with the simultaneous decrease in circulating adiponectin, an adipocyte-derived protein with insulin sensitizing, anti-inflammatory, and antiatherogenic properties [66]. Nevertheless, the consumption of a high-fat diet together with vegetable foods rich in natural antioxidants largely mitigates the negative effects on endothelial function [67]. In this study, MetS partially mediated the association between the HFADP and HFIDP and the risk of CAS. Furthermore, mechanistic studies have shown that the occurrence of MetS is closely related to leptin and adiponectin, which are metabolic inflammatory factors, and that an HFADP and HFIDP may also be associated with the risk of CAS through metabolic inflammatory mechanisms [68,69,70]. Collectively, our findings suggest that limiting the intake of an HFADP and increasing anti-inflammatory dietary patterns may reduce the risk of CAS by improving inflammation levels.

Unhealthy lifestyles are risk factors for CAS [71]. We found that maintaining ≥ 3 healthy lifestyle behaviors could further promote the protective effects of a higher DDS on CAS. In addition, the participants with unhealthy lifestyles were more inclined to adhere to an unhealthy diet than those with healthy lifestyles. Therefore, we speculate that a high adherence to ≥3 healthy lifestyle behaviors and a higher DDS magnify one another’s beneficial effects. These findings may help identify high-risk individuals with normal carotid arteries for early intervention and treatment.

A healthy diet is aimed at protecting against multiple chronic diseases simultaneously, not merely targeting a specific disease. The World Health Organization has updated its previous recommendations, and recent guidelines state that a varied diet is needed to meet the requirements for energy, protein, vitamins, and minerals, mainly through a plant-based diet, while balancing energy intake and expenditure [72]. This is similar to our findings, which primarily focused on a varied diet, plant-based foods, and the limited consumption of high-fat and processed foods. The study population was derived from the capital region of China, where the per capita gross domestic product has reached a level comparable with that of upper-middle-income countries, and the educational level of this population is relatively high. However, the median intake of vegetables in this population was 298 g/day, meaning that more than half of the individuals did not meet the recommended intake of vegetables (300–500 g/d). The median scores for the DDS and PURE were only 2 and 1, indicating insufficient public awareness and emphasis on healthy eating. Therefore, in the context of insufficient vegetable intake and dietary diversity, considering the advantages of a high-fiber diet in cardiovascular protection, it is necessary to strengthen policy implementation and promote dietary recommendations related to high fiber and diversity at the national level to gradually change the existing dietary framework.

The major strengths of the present study include its large sample size, prospective design, and diverse dietary patterns, which comprehensively demonstrated the association between different dietary pattern scores and CAS. However, we acknowledge that the present study had limitations. First, the dietary intake was assessed using an FFQ, which is subject to measurement error and recall bias. During follow-up, relying solely on the baseline dietary information obtained from the FFQs, changes in dietary habits probably distorted the observed association. Therefore, we will further consider collecting multiple 24 h dietary recall records from participants to improve the accuracy of dietary assessment. Second, the relatively short follow-up period precluded us from establishing definite causal associations between diet and CAS assessment, and the observational study could not eliminate the influence of residual confounding factors. Third, the participants included in the current analysis were Chinese adults and the dietary patterns were identified only in this population. Further studies aimed at determining the extent of the extrapolation of these results to other populations are warranted.

## 8. Conclusions

In summary, among the eight dietary patterns examined in the present study, a high adherence to an HFIDP and low adherence to an HFADP may confer the greatest risk reduction for the onset of early- and advanced-stage phenotypes of CAS. Therefore, prioritizing more plant-based foods and fewer high-fat and processed foods, and incorporating a variety of natural and anti-inflammatory foods in moderate amounts may provide an effective strategy for preventing the occurrence of CAS. Moreover, our study suggests that the modulation of metabolic inflammation may be a potential mechanism linking dietary patterns with CAS, as well as the importance of dietary diversity. Further studies replicating our findings, confirming causal relationships, and examining the detailed inflammatory mechanisms through which diet is associated with the risk for CAS are warranted.

## Figures and Tables

**Figure 1 nutrients-16-02338-f001:**
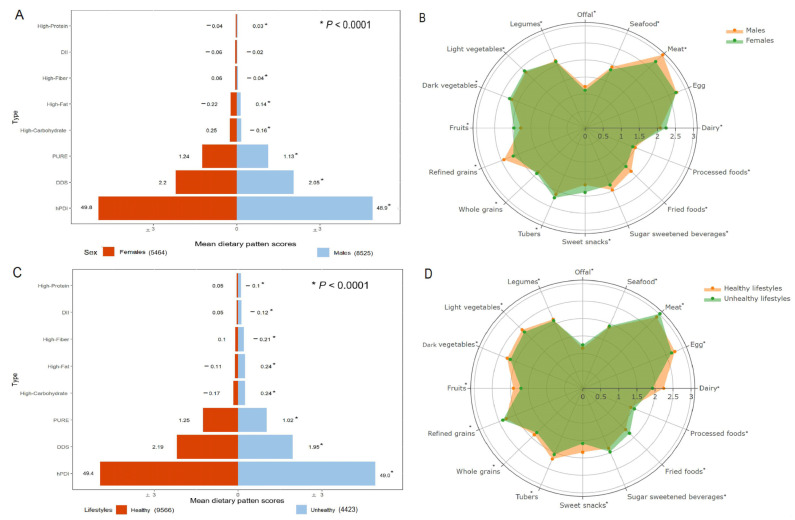
Means of eight dietary pattern scores and major food group intakes according to sex and healthy lifestyle status (*n* = 13,989). (**A**,**C**) *p*-values were calculated using linear regression. (**B**,**D**) For clarity, the radar map shows intake levels by sex and healthy lifestyle status for each food group, with light vegetables, dark vegetables, fruits, and refined grains expressed in servings per day and other food groups in servings per week. * *p* < 0.05, calculated using linear regression.

**Figure 2 nutrients-16-02338-f002:**
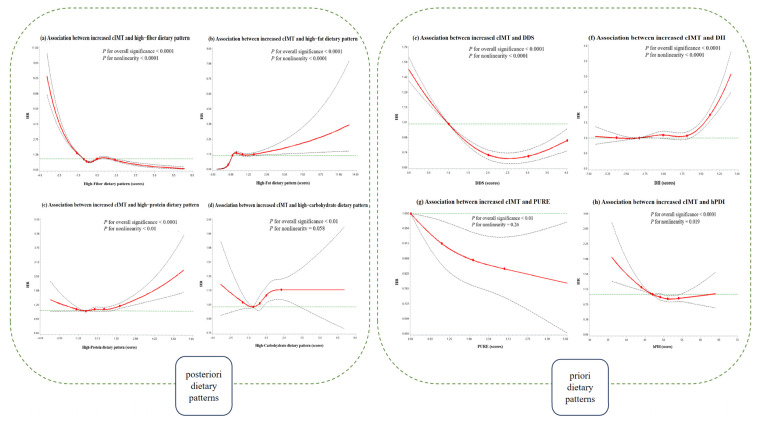
The restricted cubic spline for the association between various dietary pattern scores and increased cIMT in the whole population. Solid red lines and dashed black lines represent HR and 95% CI based on restricted cubic splines in the Cox regression model. The horizontal green dashed line represents the reference value. Knots were placed at the 5th, 25th, 50th, 75th, and 95th percentiles of the dietary pattern score distribution, and the reference value was set at the 25th percentile. Adjustment factors were age; sex; ethnicity; educational attainment; marital status; household income; physical activity; smoking status; drinking status; multivitamin supplement use; BMI; family history of cardiovascular disease; individual history of diabetes, hypertension, and dyslipidemia; total energy intake; and each other dietary pattern score (i.e., high-fiber, high-fat, high-protein, and high-carbohydrate) for a posteriori dietary patterns. Adjustment factors were age; sex; ethnicity; educational attainment; marital status; household income; physical activity; smoking status; drinking status; multivitamin supplement use; BMI; family history of cardiovascular disease; individual history of diabetes, hypertension, and dyslipidemia; and total energy intake for a priori dietary patterns. BMI: body mass index; CI: confidence interval; cIMT: carotid intima-media thickness; DDS: dietary diversity score; DII: dietary inflammation index; hPDI: healthful plant-based diet index; HR: hazard ratio; PURE: Prospective Urban Rural Epidemiology healthy diet.

**Figure 3 nutrients-16-02338-f003:**
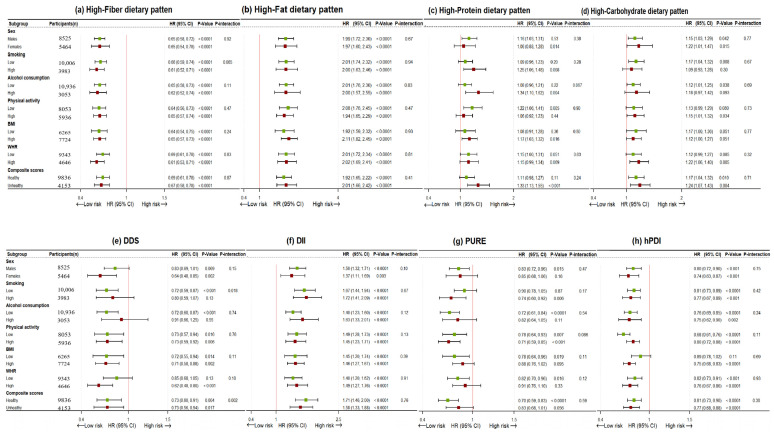
Stratified analysis of estimated associations between various dietary patterns and increased cIMT according to sex and lifestyles, by comparing the highest with the lowest quartiles. Analyses of priori and posteriori dietary patterns were adjusted for age; sex; ethnicity; educational attainment; marital status; household income; physical activity; smoking status; drinking status; multivitamin supplement use; BMI; family history of cardiovascular disease; individual history of diabetes, hypertension, and dyslipidemia; and total energy intake, with exclusion of the stratified variables as appropriate. Additionally, each other dietary pattern score was adjusted (i.e., high-fiber, high-fat, high-protein, and high-carbohydrate) for a posteriori dietary patterns. *p* for overall interaction was calculated using likelihood ratio test. BMI: body mass index; CI: confidence interval; cIMT: carotid intima-media thickness; DDS dietary diversity score; DII: dietary inflammation index; hPDI: healthful plant-based diet index; HR: hazard ratio; PURE: Prospective Urban Rural Epidemiology healthy diet; WHR: waist-to-hip ratio.

**Table 1 nutrients-16-02338-t001:** Age-adjusted baseline characteristics of participants according to categories of various dietary pattern scores ^a^.

Characteristics	Total	DDS		DII		PURE		hPDI		High-Fiber		High-Fat		High-Protein		High-Carbohydrate	
		Q1	Q4	Q1	Q4	Q1	Q4	Q1	Q4	Q1	Q4	Q1	Q4	Q1	Q4	Q1	Q4
No. of subjects	13,989	2925	737	3560	3462	4431	1704	3945	3363	3497	3498	3497	3497	3498	3497	3498	3497
Median diet score		1.00	4.00	−3.45	3.48	0.00	3.00	46.0	52.0	−0.76	1.27	−0.97	1.10	−1.06	1.10	−1.03	0.99
Age (y)	41.7 (41.5, 41.8)	41.8 (41.5, 42.1)	42.2 (41.6, 42.8)	41.9 (41.7, 42.2)	41.4 (41.1, 41.7)	41.3 (41.1, 41.6)	41.5 (41.1, 42.0)	41.0 (40.8, 41.3)	42.5 (42.3, 42.8)	41.9 (41.6, 42.2)	41.7 (41.5, 42.0)	42.9 (42.6, 43.2)	39.8 (39.5, 40.0)	42.0 (41.8, 42.3)	41.4 (41.1, 41.7)	40.6 (40.3, 40.9)	42.9 (42.6, 43.2)
Sex (Male, %)	8525 (60.9)	2,113 (72.2)	4197 (57.9)	2132 (59.9)	2145 (62.0)	2856 (64.5)	974 (57.2)	2689 (68.2)	1721 (51.2)	2241 (64.1)	2043 (58.4)	1686 (48.2)	2520 (72.1)	2060 (58.9)	2176 (62.2)	2675 (76.5)	1661 (47.5)
Ethnic group (Han, %)	5657 (40.4)	1254 (42.9)	304 (41.2)	1411 (39.6)	1455 (42.0)	1949 (44.0)	638 (37.4)	1567 (39.7)	1430 (42.5)	1449 (41.4)	1378 (39.4)	1366 (39.1)	1504 (43.0)	1601 (45.8)	1309 (37.4)	1434 (41.0)	1418 (40.5)
Educational attainment (>12 y, %)	12,895 (92.2)	2591 (88.6)	675 (91.6)	3291 (92.4)	3133 (90.5)	4030 (91.0)	1592 (93.4)	3711 (94.1)	2998 (89.2)	3207 (91.7)	3223 (92.1)	3217 (92.0)	3210 (91.8)	3097 (88.5)	3320 (94.9)	3183 (91.0)	3226 (92.2)
Currently married (%)	12,683 (90.7)	2647 (90.5)	668 (90.6)	3238 (91.0)	3125 (90.3)	4040 (91.2)	1508 (88.5)	3552 (90.0)	3044 (90.5)	3111 (88.9)	3167 (90.5)	3229 (92.3)	3081 (88.1)	3208 (91.7)	3154 (90.2)	3182 (91.0)	3172 (90.7)
Annual income (≥150,000 yuan, %)	6642 (47.5)	1209 (41.3)	408 (55.4)	1880 (52.8)	1382 (39.9)	1896 (42.8)	930 (54.6)	2048 (51.9)	1412 (42.0)	1481 (42.3)	1807 (51.7)	1597 (45.7)	1735 (49.6)	1354 (38.7)	1896 (54.2)	1681 (48.1)	1615 (46.2)
Hypertension (%)	1963 (14.0)	500 (17.1)	103 (14.0)	489 (13.7)	507 (14.6)	659 (14.9)	227 (13.3)	540 (13.7)	492 (14.6)	563 (16.1)	509 (14.6)	469 (13.4)	435 (12.4)	542 (15.5)	447 (12.8)	483 (13.8)	502 (14.4)
Diabetes (%)	665 (4.75)	231 (7.90)	53 (7.19)	180 (5.06)	176 (5.08)	223 (5.00)	80 (4.69)	194 (4.92)	154 (4.58)	199 (5.69)	191 (5.46)	175 (5.00)	154 (4.40)	173 (4.95)	169 (4.83)	192 (5.49)	159 (4.55)
Dyslipidemia (%)	6896 (49.3)	1557 (53.2)	364 (49.4)	1711 (48.1)	1763 (50.9)	2230 (50.3)	787 (46.2)	2046 (51.9)	1540 (45.8)	1854 (53.0)	1675 (47.9)	1679 (48.0)	1736 (49.6)	1751 (50.1)	1717 (49.1)	1734 (49.6)	1738 (49.7)
MetS (%)	3850 (27.5)	988 (33.8)	201 (27.3)	907 (25.5)	1008 (29.1)	1361 (30.7)	422 (24.8)	1087 (27.6)	893 (26.6)	1045 (29.9)	901 (25.8)	891 (25.5)	1008 (28.8)	1093 (31.2)	866 (24.8)	971 (27.8)	936 (26.8)
Healthy lifestyle characteristics ^b^																	
Noncurrent smoker (%)	10,006 (71.5)	1702 (58.2)	565 (76.7)	2686 (75.5)	2274 (65.7)	2925 (66.0)	1320 (77.5)	2661 (67.5)	2.601 (77.3)	2328 (66.6)	2589 (74.0)	2621 (74.9)	2377 (68.0)	2372 (67.8)	2628 (75.1)	2436 (69.6)	2625 (75.0)
Limited alcohol consumption (%)	10,936 (78.2)	2004 (68.5)	593 (80.5)	2756 (77.4)	2720 (78.6)	3345 (75.7)	1361 (79.9)	2914 (73.9)	2787 (82.9)	2688 (76.8)	2753 (78.7)	2849 (81.5)	2562 (73.2)	2676 (76.5)	2795 (79.9)	2654 (75.9)	2834 (81.0)
Regular PA (%)	8053 (57.6)	933 (31.9)	396 (53.7)	1788 (50.2)	1173 (33.9)	1603 (36.2)	877 (51.5)	1655 (42.0)	1466 (43.6)	1342 (38.4)	1720 (49.2)	1651 (47.2)	1311 (37.5)	1275 (36.5)	1713 (49.0)	1264 (36.1)	1756 (50.2)
Moderate BMI (%)	6264 (44.8)	1169 (40.0)	337 (45.7)	1611 (45.3)	1545 (44.6)	1908 (43.1)	776 (45.5)	1707 (43.3)	1565 (46.5)	1558 (44.5)	1592 (45.5)	1666 (47.6)	1459 (41.7)	1501 (42.9)	1619 (46.3)	1552 (44.4)	1568 (44.8)
Moderate WHR (%)	9343 (66.8)	1724 (58.9)	503 (68.3)	2484 (69.8)	2197 (63.5)	2793 (63.0)	1209 (71.0)	2581 (65.4)	2295 (68.2)	2235 (63.9)	2417 (69.1)	2363 (67.6)	2312 (66.1)	2204 (63.0)	2439 (69.7)	2277 (65.1)	2394 (68.4)
Family history of CVD (%)	3569 (25.5)	768 (26.3)	175 (23.7)	905 (25.4)	874 (25.2)	1074 (24.2)	474 (27.8)	1063 (27.0)	788 (23.4)	928 (26.5)	885 (25.3)	851 (24.3)	974 (27.8)	775 (22.2)	965 (27.6)	916 (26.2)	874 (25.0)
Total energy intake, kcal/d	1927.3 (1919.6, 1935.0)	1688.6 (1675.1, 1702.2)	2443.9 (2405.1, 2483.2)	2407.1 (2393.1, 2421.2)	1547.7 (1538.6, 1556.9)	1642.8 (1633.3, 1652.5)	2472.0 (2448.8, 2495.5)	2057.3 (2042.2, 2072.6)	1872.4 (1857.4, 1887.4)	1731.7 (1719.5, 1744.0)	2287.5 (2271.4, 2303.8)	1708.8 (1696.5, 1721.1)	2269.1 (2252.8, 2285.5)	1677.8 (1665.8, 1689.9)	2238 (2222.0, 2254.2)	1915.2 (1900.6, 1929.9)	2162.8 (2146.3, 2179.4)
Sugar-sweetened beverages, g/d	45.6 (44.8, 46.4)	44.4 (42.7, 46.2)	48.7 (45.1, 52.7)	46.9 (45.2, 48.5)	42.6 (41.1, 44.2)	43.2 (41.8, 44.6)	49.2 (46.8, 51.8)	53.6 (51.8, 55.4)	38.5 (37.1, 39.9)	51.2 (49.4, 53.1)	44.7 (43.1, 46.3)	23.8 (23.0, 24.5)	91.7 (88.9, 94.7)	43.9 (42.4, 45.5)	46.2 (44.6, 47.9)	38.6 (37.2, 40.0)	53.6 (51.8, 55.6)
Vegetable intake, g/d	337.1 (335.4, 338.7)	285.4 (282.5, 288.3)	393.6 (385.7, 401.7)	442.7 (439.3, 446.2)	273.4 (271.3, 275.6)	283.7 (281.7, 285.8)	454.8 (449.5, 460.2)	307.3 (304.6, 310.0)	386.1 (382.4, 389.8)	266.9 (265.3, 268.5)	494.4 (491.4, 497.3)	353.4 (350.0, 356.9)	334.5 (331.3, 337.8)	334.3 (331.1, 337.6)	343.9 (340.6, 347.3)	353.1 (349.7, 356.6)	339.6 (336.3, 342.9)
Fruit intake, g/d	190.5 (188.8, 192.2)	88.5 (87.3, 89.7)	256.3 (249.6, 263.2)	253.2 (249.2, 257.3)	130.4 (128.3, 132.5)	157.2 (154.9, 159.6)	272.2 (265.6, 278.9)	173.5 (170.6, 176.4)	219.8 (215.9, 223.8)	137.3 (135.1, 139.6)	230.4 (226.6, 234.3)	199.8 (196.2, 203.4)	186.7 (183.4, 190.1)	163.2 (160.4, 166.1)	219.7 (215.8, 223.6)	161.5 (158.7, 164.4)	227.8 (223.9, 231.9)
Whole grain intake, g/d	31.1 (30.6, 31.5)	23.1 (22.41, 23.81)	44.6 (42.0, 47.3)	57.8 (56.4,59.2)	16.9 (16.4, 17.3)	24.6 (24.0, 25.2)	46.0 (44.3, 47.9)	25.5 (24.8, 26.1)	41.4 (40.2, 42.6)	23.2 (22.6, 23.8)	42.1 (41.0, 43.3)	27.7 (26.9, 28.4)	35.4 (34.5, 36.5)	29.2 (28.4, 30.1)	34.4 (33.4, 35.4)	18.5 (18.0, 18.9)	65.6 (64.1, 67.2)
Multivitamin supplement use (%)	3400 (24.3)	530 (18.1)	251 (34.1)	1019 (28.6)	672 (19.4)	850 (19.2)	561 (32.9)	1006 (25.5)	819 (24.4)	819 (23.4)	935 (26.7)	899 (25.7)	832 (23.8)	699 (20.0)	1033 (29.5)	711 (20.3)	1032 (29.5)

Abbreviations: Q1 and Q4: the lowest and highest quartiles or groups of dietary pattern scores; BMI: body mass index; CVD: cardiovascular disease; DDS: dietary diversity score; DII: dietary inflammation index; hPDI: healthful plant-based diet index; MetS: metabolic syndrome; PA: physical activity; PURE: Prospective Urban Rural Epidemiology healthy diet; WHR: waist-to-hip ratio. ^a^ Data are presented as age-adjusted mean (95% CI) for continuous variables and percentages (%) for categorical variables. We only display the result of Q1 and Q4 of dietary pattern scores for simplicity. ^b^ The definition of healthy lifestyle characteristics is described in Appendix A.

**Table 2 nutrients-16-02338-t002:** Nutritional profiles by diet score category in the whole population.

Food Groups	Factor Loadings				DDS ^a^		DII ^a^		PURE ^a^		hPDI ^a^	
	High-Fiber	High-Fat	High-Protein	High-Carbohydrate	Q1 (Least Healthy)	Q4 (Most Healthy)	Q1 (Most Healthy)	Q4 (Least Healthy)	Q1 (Least Healthy)	Q4 (Most Healthy)	Q1 (Least Healthy)	Q4 (Most Healthy)
Dairy products, g per day	0.01	−0.17	**0.62**	0.28	70.5	176.5	124.6	77.1	39.6	173.2	155.4	62.0
Eggs, g per day	0.07	−0.04	**0.67**	0.009	21.1	53.7	37.5	24.5	25.7	39.2	41.4	23.4
Meat, g per day	0.05	0.31	**0.54**	−0.25	48.3	64.2	58.0	44.3	46.2	59.9	66.1	38.5
Seafood, g per day	0.22	0.21	**0.45**	−0.04	20.1	43.8	35.0	16.3	18.3	43.0	32.0	20.6
Offal, g per day	0.09	**0.51**	0.11	−0.07	11.6	15.9	15.8	8.49	11.0	16.1	15.8	9.43
Legumes, g per day	**0.33**	0.21	0.26	0.08	49.7	118.4	91.0	35.7	42.3	100.8	57.5	67.0
Light vegetables, g per day	**0.87**	−0.04	0.03	−0.03	157.0	214.2	236.0	144.2	145.6	244.2	164.2	201.7
Dark vegetables, g per day	**0.87**	−0.04	0.03	−0.03	145.9	197.2	222.5	138.0	143.1	224.0	155.6	188.8
Fruits, g per day	**0.34**	−0.07	0.19	0.24	95.3	268.2	273.8	151.5	178.4	302.6	197.5	237.5
Refined grains, g per day	0.10	0.16	0.11	**−0.70**	238.3	216.6	230.5	225.6	231.5	231.6	262.8	202.8
Whole grains, g per day	0.23	0.10	0.03	**0.70**	34.6	64.0	76.3	21.4	35.2	62.0	36.0	55.5
Tubers, g per day ^b^	**0.40**	0.10	0.10	0.31	38.1	59.6	62.1	33.7	39.0	60.8	39.9	52.4
Sweet snacks, g per day ^e^	0.04	0.22	0.27	**0.34**	37.5	60.5	57.2	37.2	36.5	62.6	57.3	42.0
Sugar-sweetened beverages, g per day ^f^	−0.03	**0.48**	0.03	0.09	85.1	97.8	89.6	80.8	78.8	93.7	99.9	73.5
Fried foods, g per day ^d^	−0.07	**0.71**	0.004	−0.02	49.8	49.8	49.3	45.9	46.9	50.4	54.1	42.9
Processed foods, g per day ^c^	0.04	**0.60**	−0.008	0.003	7.11	8.13	8.44	6.51	7.17	8.44	10.4	5.37
Explained variation in food groups (%)	15.6	10.9	8.14	7.18	-	-	-	-	-	-	-	-
Alcohol, g per day ^g^	10.3	10.3	10.3	10.3	16.3	9.86	11.2	10.2	11.6	9.33	12.3	8.40
Cholesterol, mg per day	547.9	547.9	547.9	547.9	456.2	805.8	677.7	409.1	461.2	717.1	710.3	419.7
Carbohydrates, %E	63.7	63.7	63.7	63.7	62.7	59.9	62.7	64.6	64.9	61.9	61.9	65.3
Fats, %E	22.6	22.6	22.6	22.6	22.7	23.9	21.8	23.0	22.1	22.7	25.0	20.5
Saturated, %E	7.91	7.91	7.91	7.91	7.90	8.49	7.48	8.16	7.52	7.99	9.15	6.82
Monounsaturated, %E	7.95	7.95	7.95	7.95	7.98	8.25	7.60	8.13	7.86	7.87	8.85	7.13
Polyunsaturated, %E	5.85	5.85	5.85	5.85	5.85	6.24	5.87	5.75	5.78	5.98	5.96	5.73
Polyunsaturated-to-saturated fat ratio	0.76	0.76	0.76	0.76	0.76	0.77	0.81	0.72	0.78	0.77	0.66	0.85
Other, %E ^h^	0.92	0.92	0.92	0.92	0.99	0.89	0.82	1.00	0.97	0.84	1.04	0.80
Protein, %E	16.2	16.2	16.2	16.2	16.0	17.6	16.4	15.8	15.7	16.9	16.8	15.7

Abbreviations: Q1 and Q4: the lowest and highest quantiles or groups of dietary pattern scores; DDS: dietary diversity score; DII: dietary inflammation index; hPDI: healthful plant-based diet index; PURE: Prospective Urban Rural Epidemiology healthy diet; %E: % of total energy intake. The loading factors displayed in bold font in the posterior dietary patterns represent the foods that make a major contribution to the respective dietary pattern. ^a^ Table shows mean values for each food and nutrient. ^b^ Tubers include sweet potato, yam, taro, potato, and products. ^c^ Processed foods include pickles, salted fish, ham, sausage, canned food. ^d^ Fried foods include fried dough twist, deep-fried cake, fried dough sticks, and instant noodles. ^e^ Sweet snacks include jam, honey, cakes, cookies, biscuits, pastries, ice cream, confections, candies, and added sugar. ^f^ Sugar-sweetened beverages include coffee, cocoa, tea, fruit and vegetable juices, soft beverages. ^g^ Alcoholic beverages include red wine, white spirits, beer, and other alcoholic beverages. ^h^ Other is comprised of glycerol and other aldehydes.

**Table 3 nutrients-16-02338-t003:** Association of the dietary pattern scores with CAS in the whole population (*n* = 13,989).

Dietary Patterns	*n*	Increased cIMT			CP		
		Cases (%) (*n* = 3732)	Incident Rate (Events per 1000 Person-Years)	HR (95% CI)	Cases (%) (*n* = 2861)	Incident Rate (Events per 1000 Person-Years)	HR (95% CI)
High-Fiber (range) ^a^							
Q1 (−3.44, −0.58)	3497	1387 (39.7)	133.6	Ref	1068 (30.5)	97.3	Ref
Q2 (−0.58, −0.32)	3496	670 (19.2)	63.1	0.50 (0.45, 0.54)	516 (14.8)	47.0	0.51 (0.46, 0.57)
Q3 (−0.32, 0.44)	3498	760 (21.7)	72.2	0.54 (0.49, 0.59)	579 (16.6)	53.1	0.55 (0.49, 0.61)
Q4 (0.44, 7.38)	3498	915 (26.2)	86.1	0.65 (0.59, 0.71)	698 (20.0)	63.1	0.65 (0.59, 0.73)
*p* for trend				<0.0001			<0.0001
HR (95% CI) per 1 SD				0.77 (0.74, 0.80)			0.77 (0.74, 0.81)
PAR (95% CI)				0.17 (0.14, 0.20)			0.17 (0.13, 0.21)
High-Fat (range) ^a^							
Q1 (−2.51, −0.69)	3498	525 (15.0)	48.7	Ref	417 (11.9)	37.8	Ref
Q2 (−0.69, −0.20)	3497	1137 (32.5)	106.7	2.15 (1.94, 2.38)	868 (24.8)	77.4	2.00 (1.78, 2.25)
Q3 (−0.20, 0.48)	3496	1087 (31.1)	105.7	2.16 (1.94, 2.40)	837 (23.9)	77.5	2.04 (1.80, 2.30)
Q4 (0.48, 13.1)	3498	983 (28.1)	94.2	1.96 (1.75, 2.20)	739 (21.1)	68.1	1.84 (1.61, 2.10)
*p* for trend				<0.0001			<0.0001
HR (95% CI) per 1 SD				1.13 (1.10, 1.17)			1.10 (1.06, 1.15)
PAR (95% CI)				0.48 (0.42, 0.53)			0.44 (0.38, 0.50)
High-Protein (range) ^a^							
Q1 (−3.29, −0.69)	3498	911 (26.0)	85.1	Ref	691 (19.8)	62.1	Ref
Q2 (−0.69, −0.10)	3497	904 (25.9)	86.5	1.02 (0.93, 1.12)	701 (20.1)	64.4	1.02 (0.92, 1.13)
Q3 (−0.10, 0.57)	3496	884 (25.3)	83.6	1.03 (0.94, 1.14)	689 (19.7)	64.0	1.06 (0.95, 1.18)
Q4 (0.57, 6.40)	3498	1033 (29.5)	99.1	1.13 (1.02, 1.25)	780 (22.3)	71.8	1.12 (0.99, 1.25)
*p* for trend				0.040			0.087
HR (95% CI) per 1 SD				1.04 (1.00, 1.08)			1.03 (1.00, 1.08)
PAR (95% CI)				0.04 (0.01, 0.07)			0.04 (0.003, 0.07)
High-Carbohydrate (range) ^a^							
Q1 (−3.32, −0.61)	3498	924 (24.8)	86.7	Ref	696 (19.9)	62.7	Ref
Q2 (−0.61, −0.05)	3497	895 (25.1)	85.3	1.05 (0.96, 1.15)	678 (19.4)	62.1	1.05 (0.95, 1.17)
Q3 (−0.05, 0.49)	3496	910 (26.4)	85.2	1.05 (0.96, 1.16)	694 (19.9)	62.6	1.05 (0.94, 1.17)
Q4 (0.49, 7.04)	3498	1003 (28.7)	97.2	1.17 (1.07, 1.29)	793 (22.7)	73.4	1.20 (1.08, 1.34)
*p* for trend				0.009			0.011
HR (95% CI) per 1 SD				1.03 (1.00, 1.07)			1.04 (1.00, 1.08)
PAR (95% CI)				0.05 (0.02, 0.08)			0.05 (0.02, 0.08)
DDS (range) ^b^							
Q1 [0.00, 1.00]	2925	986 (33.7)	110.1	Ref	754 (25.8)	80.6	Ref
Q2 [2.00]	7244	1743 (24.1)	78.9	0.75 (0.69, 0.81)	1333 (18.4)	58.0	0.76 (0.69, 0.83)
Q3 [3.00]	3083	807 (26.2)	89.9	0.78 (0.70, 0.86)	624 (20.2)	66.9	0.81 (0.72, 0.91)
Q4 [4.00, 7.00]	737	196 (26.6)	92.5	0.74 (0.63, 0.87)	150 (20.4)	67.5	0.72 (0.59, 0.87)
*p* for trend				<0.0001			<0.001
HR (95% CI) per 1 SD				0.89 (0.85, 0.92)			0.90 (0.86, 0.93)
PAR (95% CI)				0.07 (0.03, 0.10)			0.07 (0.03, 0.11)
DII (range) ^b^							
Q1 (−6.39, −2.26)	3560	930 (26.1)	86.7	Ref	701 (19.7)	62.9	Ref
Q2 (−2.26, −0.01)	3477	892 (25.7)	84.3	1.13 (1.03, 1.25)	688 (19.8)	62.6	1.14 (1.02, 1.27)
Q3 (−0.01, 2.22)	3490	872 (25.0)	83.7	1.22 (1.09, 1.35)	695 (19.9)	64.0	1.25 (1.11, 1.41)
Q4 (2.22, 6.37)	3462	1038 (30.0)	99.5	1.66 (1.48, 1.87)	777 (22.4)	71.3	1.59 (1.39, 1.82)
*p* for trend				<0.0001			<0.0001
HR (95% CI) per 1 SD				1.24 (1.19, 1.30)			1.22 (1.16, 1.29)
PAR (95% CI)				0.13 (0.06, 0.19)			0.14 (0.06, 0.21)
PURE (range) ^b^							
Q1 [0]	4431	1157 (26.1)	90.1	Ref	881 (19.9)	65.8	Ref
Q2 [1.0]	4902	1331 (27.2)	88.3	0.86 (0.79, 0.93)	1026 (20.9)	65.2	0.87 (0.79, 0.95)
Q3 [2.0]	2952	784 (26.6)	87.1	0.77 (0.70, 0.85)	606 (20.5)	65.1	0.80 (0.71, 0.90)
Q4 [3.0, 5.0]	1704	460 (27.0)	88.0	0.76 (0.67, 0.87)	348 (20.4)	63.5	0.75 (0.65, 0.87)
*p* for trend				<0.0001			<0.0001
HR (95% CI) per 1 SD				0.90 (0.87, 0.94)			0.90 (0.86, 0.95)
PAR (95% CI)				0.05 (0.01, 0.08)			0.05 (0.02, 0.09)
hPDI (range) ^b^							
Q1 [36.0, 47.0]	3945	1208 (30.6)	101.2	Ref	923 (23.4)	74.0	Ref
Q2 (47.0, 49.0]	3408	874 (25.7)	85.5	0.88 (0.81, 0.96)	653 (19.2)	61.2	0.85 (0.76, 0.94)
Q3 (49.0, 51.0]	3273	842 (25.7)	87.3	0.87 (0.80, 0.96)	667 (20.4)	66.4	0.90 (0.82, 0.99)
Q4 (51.0, 64.0]	3363	808 (24.0)	78.1	0.78 (0.71, 0.85)	618 (18.4)	57.6	0.79 (0.72, 0.88)
*p* for trend				<0.0001			<0.0001
HR (95% CI) per 1 SD				0.91 (0.88, 0.94)			0.92 (0.89, 0.96)
PAR (95% CI)				0.06 (0.04, 0.09)			0.06 (0.03, 0.09)

Abbreviations: Q1 and Q4: the lowest and highest quartiles or groups of dietary pattern scores; BMI: body mass index; CAS: carotid atherosclerosis; CI: confidence interval; cIMT: carotid intima-media thickness; CP: carotid plaque; DDS dietary diversity score; DII: dietary inflammation index; hPDI: healthful plant-based diet index; HR: hazard ratio; PAR: population attributable risk; PURE: Prospective Urban Rural Epidemiology healthy diet. ^a^ Multivariable models are adjusted for age; sex; ethnicity; educational attainment; marital status; household income; physical activity; smoking status; drinking status; multivitamin supplement use; BMI; family history of cardiovascular disease; individual history of diabetes, hypertension, and dyslipidemia; total energy intake; and each other dietary pattern score (i.e., high-fiber, high-fat, high-protein, and high-carbohydrate). ^b^ Multivariable models are adjusted for age; sex; ethnicity; educational attainment; marital status; household income; physical activity; smoking status; drinking status; multivitamin supplement use; BMI; family history of cardiovascular disease; individual history of diabetes, hypertension, and dyslipidemia; and total energy intake.

**Table 4 nutrients-16-02338-t004:** Comparison of associations between various dietary pattern scores with CAS (*n* = 13,989).

AUC (95% CI)								
	High-Fiber Dietary Pattern ^c, d, e, f, g, h^	High-Fat Dietary Pattern ^c, d, e, f, g, h^	High-Protein Dietary Pattern ^a, b^	High-Carbohydrate Dietary Pattern ^a, b^	DDS ^a, b^	DII ^a, b^	PURE ^a, b^	hPDI ^a, b^
Increased cIMT	0.56 (0.55, 0.57)	0.55 (0.54, 0.56)	0.51 (0.50, 0.52)	0.52 (0.50, 0.53)	0.52 (0.51, 0.53)	0.52 (0.51, 0.53)	0.51 (0.50, 0.52)	0.52 (0.51, 0.53)
CP	0.56 (0.55, 0.57)	0.54 (0.53, 0.55)	0.51 (0.50, 0.52)	0.52 (0.50, 0.53)	0.52 (0.51, 0.53)	0.52 (0.51, 0.53)	0.51 (0.50, 0.52)	0.52 (0.51, 0.52)

Abbreviations: AUC: area under the curve; CAS: carotid atherosclerosis; CI: confidence interval; cIMT: carotid intima-media thickness; CP: carotid plaque; DDS dietary diversity score; DII: dietary inflammation index; hPDI: healthful plant-based diet index; Prospective Urban Rural Epidemiology. ^a^ *p* < 0.05 for ROC contrast estimation of this score vs. high-fiber dietary pattern diet score. ^b^ *p* < 0.05 for ROC contrast estimation of this score vs. high-fat dietary pattern diet score. ^c^ *p* < 0.05 for ROC contrast estimation of this score vs. high-protein dietary pattern diet score. ^d^ *p* < 0.05 for ROC contrast estimation of this score vs. high-carbohydrate dietary pattern diet score. ^e^ *p* < 0.05 for ROC contrast estimation of this score vs. DDS score. ^f^ *p* < 0.05 for ROC contrast estimation of this score vs. DII score. ^g^ *p* < 0.05 for ROC contrast estimation of this score vs. PURE score. ^h^ *p* < 0.05 for ROC contrast estimation of this score vs. hPDI score.

**Table 5 nutrients-16-02338-t005:** Mediation analysis of the association between dietary patterns and CAS mediated by metabolic syndrome.

Dietary Patterns	CAS	No. of MetS	Incident Rate (Events per 1000 Person-Years)	HR (95% CI) ^NIE^	HR (95% CI) ^NDE a^	HR (95% CI) ^TE^	PM (%) ^b^	*p*-Value
High-Fiber	Increased cIMT	3850	88.5	0.993 (0.988, 0.998)	0.851 (0.824, 0.879)	0.845 (0.818, 0.873)	3.70	0.011
High-Fat	Increased cIMT	3850	88.5	1.007 (1.002, 1.012)	1.168 (1.131, 1.207)	1.176 (1.138, 1.215)	4.41	0.011
High-Fiber	CP	3850	65.2	0.993 (0.988, 0.998)	0.853 (0.823, 0.885)	0.848 (0.817, 0.880)	3.88	0.011
High-Fat	CP	3850	65.2	1.007 (1.002, 1.013)	1.143 (1.101, 1.013)	1.151 (1.108, 1.195)	5.38	0.011

Abbreviations: CAS: carotid atherosclerosis; CI: confidence interval; cIMT: carotid intima-media thickness; CP: carotid plaque; HR^NIE^: hazard ratio for the natural indirect effect; HR^NDE^: hazard ratio for the natural direct effects; HR^TE^: hazard ratio for the total effects; PM: proportion of mediation. Adjustment factors were age; sex; ethnicity; educational attainment; marital status; household income; physical activity; smoking status; drinking status; multivitamin supplement use; BMI; family history of cardiovascular disease; individual history of diabetes, hypertension, and dyslipidemia; total energy intake; and each other dietary pattern score (i.e., high-fiber, high-fat, high-protein, and high-carbohydrate). ^a^ The calculation of natural direct effects needs to be conditional on the level of the covariates. ^b^ Proportion of mediation = natural indirect effect/(natural direct effect + natural indirect effect) × 100.

## Data Availability

The data used to support the findings of this study are not publicly available since the study involved individual check-up data but are available from the corresponding author on reasonable request.

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
