# Peer review of "Adherence to a Healthy Diet and Risk of Multiple Carotid Atherosclerosis Subtypes: Insights from the China MJ Health Check-Up Cohort"

_nutrients, 2024, doi:10.3390/nu16142338_

Round 1

Reviewer 1 Report

Comments and Suggestions for Authors

The main purpose of the research is to assess the associations between various dietary pattern scores and the onset of different subtypes of carotid atherosclerosis (CAS) in adults. Specifically, the study aims to compare the performance of eight dietary pattern scores, including four a priori dietary patterns (Dietary Diversity Score (DDS), Dietary Inflammatory Index (DII), Prospective Urban Rural Epidemiology (PURE), and healthful Plant-based Diet Index (hPDI)) and four a posteriori dietary patterns derived from the MJ dietary data. The research seeks to provide relevant dietary guidance for Chinese populations based on these associations and to identify optimal dietary patterns for preventing CAS.

Here are my comments regarding this paper:

- The paper acknowledges that the dietary intake assessment using a Food Frequency Questionnaire (FFQ) is subject to measurement error and recall bias. Implementing a more robust method for dietary assessment, such as multiple 24-hour dietary recalls or food diaries over time, could have enhanced accuracy.

- Explain more thoroughly the detailed inflammatory mechanisms through which diet is associated with CAS. This could involve exploring specific biochemical pathways and the role of various nutrients in inflammation and atherosclerosis development.

- Enhance the discussion on policy implementation and public health recommendations based on the study findings. Highlighting how the results can inform dietary guidelines and public health strategies can increase the paper's practical impact.

- Outline specific areas for future research, such as the exploration of other dietary patterns not covered in the current study or the examination of dietary impacts on other cardiovascular outcomes

- Expand the literature about how certain dietary behaviours can have a positive impact on cardiovascular health in general, for example considering the impact COVID-19 had on the Chinese population's health.

This paper, for instance 10.5603/CJ.a2021.0159 stresses how the importance of maintaining cardiovascular health is underscored by the complications observed in COVID-19 survivors, such as pulmonary arterial hypertension and right ventricular systolic dysfunction, which can potentially be mitigated by healthy dietary patterns​. Moreover, both papers discuss the role of inflammation in cardiovascular health. The paper under review explores how dietary patterns can influence inflammation and subsequently carotid atherosclerosis, while Rossi et al. describe the inflammatory impacts on the cardiovascular system in COVID-19 survivors.

Incorporating these improvements can enhance the rigor, validity, and impact of the research, making it more valuable to the scientific community and public health policymakers​.

Author Response

Reviewer #1:

  1. The paper acknowledges that the dietary intake assessment using a Food Frequency Questionnaire (FFQ) is subject to measurement error and recall bias. Implementing a more robust method for dietary assessment, such as multiple 24-hour dietary recalls or food diaries over time, could have enhanced accuracy.

Response:

Thank you for this suggestion. In the subsequent data collection, we will further consider collecting multiple 24-hour dietary recall records from participants to improve the accuracy of dietary assessment. In lines 141-142 of the discussion section we added:

" We will further consider collecting multiple 24-hour dietary recall records from participants to improve the accuracy of dietary assessment. "

  1. Explain more thoroughly the detailed inflammatory mechanisms through which diet is associated with CAS. This could involve exploring specific biochemical pathways and the role of various nutrients in inflammation and atherosclerosis development.

Response:

    Thank you for your suggestion. We have added detailed description of the inflammatory mechanisms related to diet and CAS in lines 89-103 of the discussion section. The revised content is as follows:

" Such high-density and low-quality foods not only contain excessive salt, saturated fat, and refined sugars, but also add or generate trans-fatty acids during processing, which may cause an activation of the innate immune system, most likely by an excessive production of pro-inflammatory cytokines associated with a reduced production of anti-inflammatory cytokines [47-49]. In addition, a high-fat diet produces endothelial activation, as evidenced by increased concentrations of the adhesion molecules VCAM-1 (vascular cell adhesion molecule-1) and ICAM-1 (intercellular adhesion molecule-1), in association with raised plasma concentrations of interleukin-6 (IL-6) and tumor necrosis factor-a (TNF-a) [50]. The high-fat diet also increases the circulating levels of interleukin-18 (IL-18), a pro-inflammatory cytokine supposed to be involved in plaque destabilization, associated with the simultaneous decrease of circulating adiponectin, an adipocyte-derived protein with insulin sensitizing, anti-inflammatory, and antiatherogenic properties [51]. Nevertheless, consumption of a high-fat diet together with vegetable foods rich in natural antioxidants largely mitigates the negative effects on endothelial function [52] . "

References

  1. Menezes, C.A.; Magalhaes, L.B.; da Silva, J.T.; da Silva Lago, R.M.R.; Gomes, A.N.; Ladeia, A.M.T.; Vianna, N.A.; Oliveira, R.R. Ultra-Processed Food Consumption Is Related to Higher Trans Fatty Acids, Sugar Intake, and Micronutrient-Impaired Status in Schoolchildren of Bahia, Brazil. Nutrients 2023, 15, doi:10.3390/nu15020381.
  2. Lopez-Garcia, E.; Schulze, M.B.; Meigs, J.B.; Manson, J.E.; Rifai, N.; Stampfer, M.J.; Willett, W.C.; Hu, F.B. Consumption of trans fatty acids is related to plasma biomarkers of inflammation and endothelial dysfunction. J Nutr 2005, 135, 562-566, doi:10.1093/jn/135.3.562.
  3. Esposito, K.; Giugliano, D. Diet and inflammation: a link to metabolic and cardiovascular diseases. Eur Heart J 2006, 27, 15-20, doi:10.1093/eurheartj/ehi605.
  4. Nappo, F.; Esposito, K.; Cioffi, M.; Giugliano, G.; Molinari, A.M.; Paolisso, G.; Marfella, R.; Giugliano, D. Postprandial endothelial activation in healthy subjects and in type 2 diabetic patients: role of fat and carbohydrate meals. J Am Coll Cardiol 2002, 39, 1145-1150, doi:10.1016/s0735-1097(02)01741-2.
  5. Esposito, K.; Nappo, F.; Giugliano, F.; Di Palo, C.; Ciotola, M.; Barbieri, M.; Paolisso, G.; Giugliano, D. Meal modulation of circulating interleukin 18 and adiponectin concentrations in healthy subjects and in patients with type 2 diabetes mellitus. Am J Clin Nutr 2003, 78, 1135-1140, doi:10.1093/ajcn/78.6.1135.
  6. Esposito, K.; Nappo, F.; Giugliano, F.; Giugliano, G.; Marfella, R.; Giugliano, D. Effect of dietary antioxidants on postprandial endothelial dysfunction induced by a high-fat meal in healthy subjects. Am J Clin Nutr 2003, 77, 139-143, doi:10.1093/ajcn/77.1.139.

  1. Enhance the discussion on policy implementation and public health recommendations based on the study findings. Highlighting how the results can inform dietary guidelines and public health strategies can increase the paper's practical impact.

Response:

   Thank you for your advice. Based on the results of this study, we have added diet-related recommendations to inform the formulation of public health recommendations in the discussion section from lines 131 to 134. The added content is as follows:

"Therefore, in the context of insufficient vegetable intake and dietary diversity, considering the advantages of a high-fiber diet in cardiovascular protection, it is necessary to strengthen policy implementation and promote dietary recommendations related to high fiber and diversity at the national level to gradually change the existing dietary framework."

  1. Outline specific areas for future research, such as the exploration of other dietary patterns not covered in the current study or the examination of dietary impacts on other cardiovascular outcomes

Response:

   We thank the reviewer for this comment. We will further examine the association between dietary patterns mentioned in this study and the risk of coronary artery calcification in a large sample population.

  1. Expand the literature about how certain dietary behaviours can have a positive impact on cardiovascular health in general, for example considering the impact COVID-19 had on the Chinese population's health.

-This paper, for instance 10.5603/CJ.a2021.0159 stresses how the importance of maintaining cardiovascular health is underscored by the complications observed in COVID-19 survivors, such as pulmonary arterial hypertension and right ventricular systolic dysfunction, which can potentially be mitigated by healthy dietary patterns. Moreover, both papers discuss the role of inflammation in cardiovascular health. The paper under review explores how dietary patterns can influence inflammation and subsequently carotid atherosclerosis, while Rossi et al. describe the inflammatory impacts on the cardiovascular system in COVID-19 survivors.

Response:

Thank you for your thoughtful advice. In view of the fact that the level of inflammation in COVID-19 patients may affect the occurrence of cardiovascular diseases, previous studies have also confirmed that diet can affect the level of inflammation, suggesting that diet may play a certain role in the prevention of cardiovascular diseases in COVID-19 patients through inflammatory mechanisms. Since participant data on COVID-19 status was not collected in this study, the research does not address related aspects. However, the impact of diet on the occurrence and progression of cardiovascular diseases in a large number of COVID-19 patients will be assessed in further studies.

Reviewer 2 Report

Comments and Suggestions for Authors

The manuscript is very well written. The novelty of the study is relatively low, however the methodology applied to the research and the rigorous statistical analysis render the study of clinical interest. I have only two minor points to suggest: 1) the quality of the figures should be improved, since they appear at low resolution; 2) authors should acknowledge the fact that dietary pattern in the selected study population may not be applicable to other populations.  Beyond these minor points, I found the manuscript interesting  and well presented. 

Author Response

Reviewer #2:

  1. The quality of the figures should be improved, since they appear at low resolution

Response:

We have increased the resolution of the picture to improve the quality of the image.

  1. Authors should acknowledge the fact that dietary pattern in the selected study population may not be applicable to other populations.

Response:

We agree with the reviewer and therefore refer to this in the discussion section in lines 145-148:

"Third, participants included in the current analysis were Chinese adults and dietary patterns were identified only in this population. Further studies aimed at determining the extent of extrapolation of these results to other populations are warranted."